# National Cross-Sectional Study of Mental Health Screening Practices for Primary Caregivers of NICU Infants

**DOI:** 10.3390/children9060793

**Published:** 2022-05-28

**Authors:** Cooper Bloyd, Snehal Murthy, Clara Song, Linda S. Franck, Christina Mangurian

**Affiliations:** 1School of Medicine, University of California, San Francisco, CA 94143, USA; cooper.bloyd@ucsf.edu (C.B.); snehal.murthy@ucsf.edu (S.M.); 2Department of Pediatrics, University of California, San Francisco, CA 94143, USA; 3Southern California Permanente Medical Group, Pasadena, CA 91188, USA; clarasong@me.com; 4School of Nursing, University of California, San Francisco, CA 94143, USA; linda.franck@ucsf.edu; 5Department of Psychiatry and Behavioral Sciences, Weill Institute for Neurosciences, University of California, San Francisco, CA 94143, USA; 6Department of Epidemiology and Biostatistics, University of California, San Francisco, CA 94143, USA

**Keywords:** postpartum, mood disorder, depression, anxiety, mental health, screening, NICU

## Abstract

Universal screening for postpartum mood and anxiety disorders (PMADs) has been recommended for all new parents at outpatient postpartum and well-child visits. However, parents of newborns admitted to the NICU are rarely able to access these services during their infant’s prolonged hospitalization. The objective of this study was to determine the prevalence of mental health screening and treatment programs for parents or other primary caregivers in NICUs across the country. In this cross-sectional study, US NICU medical directors were invited to complete an online survey about current practices in mental health education, screening, and treatment for primary caregivers of preterm and ill infants in the NICU. Comparative analyses using Fisher’s exact test were performed to evaluate differences in practices among various NICU practice settings. Survey responses were obtained from 75 out of 700 potential sites (10.7%). Of participating NICUs, less than half routinely provided caregivers with psychoeducation about mental health self-care (*n* = 35, 47%) or routinely screened caregivers for PPD or other mental health disorders (*n* = 33, 44%). Nearly one-quarter of the NICUs did not provide any PMAD screening (*n* = 17, 23%). Despite consensus that postpartum psychosocial care is essential, routine mental health care of primary caregivers in the NICU remains inadequate.

## 1. Introduction

Postpartum depression (PPD) affects as many as 1-in-7 birthing mothers and 1-in-10 fathers annually in the United States (US) [1,2] and is considered to be under-diagnosed and under-treated [3,4,5]. It is a leading cause of maternal morbidity and mortality [6] and adversely affects the mental and physical well-being of infants, including suboptimal feeding and growth, poor maternal-infant bonding, and impaired cognitive, language, and socio-emotional development [7]. To improve outcomes, early universal PPD screenings and interventions have been recommended by the American Academy of Pediatrics (AAP) [8], the American College of Obstetricians and Gynecologists [9], and the US Preventive Services Task Force [10]. Screening is typically performed in the postpartum units before discharge, during outpatient maternal postpartum visits, and more recently in pediatric settings during well-baby visits [8].

However, for many primary caregivers (parents and guardians), early outpatient screening may not be possible. About 7–12% of infants delivered annually in the US begin life in a Neonatal Intensive Care Unit (NICU) [11,12,13]. Compared to the general population, primary caregivers of NICU infants are diagnosed with higher rates of postpartum mood and anxiety disorders (PMADs) including anxiety [14,15], depression [3,16,17], and post-traumatic stress [16,18]. Within the first postpartum year, 20–30% of NICU parents experience a diagnosable mental disorder, with an additional proportion experiencing subclinical symptoms [19]. Given the increased risk of PMADs in caregivers of NICU infants, early identification and intervention for this population is critical to prevent long-term sequelae for these already vulnerable families.

For newborns admitted to higher level NICUs, where lengthy hospitalizations are common, the NICU becomes the primary site of contact with the healthcare system for family members. Thus, NICU intake has been suggested as an opportune time for mental health screening and intervention for caregivers [19,20]. In 2015, the National Perinatal Association (NPA) put forth recommendations for the provision of psychosocial care for parents in the NICU, including considerations for staffing, layered support services, and screening guidelines [21]. However, our recent systematic review identified few studies describing such programs [22]. To our knowledge, universal PMAD screening and treatment of NICU parents is not currently a standard of care. Therefore, the objective of this study was to determine the prevalence of mental health education, screening, and treatment programs for parents or other primary caregivers in NICUs across the US. By examining the current status of NICU mental health services, as well as common barriers and facilitators to implementation, novel solutions to promote mental health of NICU caregivers may emerge.

## 2. Materials and Methods

### 2.1. Study Design, Sample, and Recruitment

In this cross-sectional study, U.S. NICU medical directors were invited to complete an online survey about current practices in mental health education, screening, and treatment for parents and other primary caregivers of infants in the NICU. The online survey was distributed electronically to the 700 NICU medical directors in the listserv of the American Academy of Pediatrics, Section on Neonatal-Perinatal Medicine, which represents approximately 60% of all NICUs nationally [23]. Survey data were collected and managed using REDCap (Research Electronic Data Capture; Nashville, TN, USA, 37212) electronic data capture tools hosted at the University of California, San Francisco [24]. Participation was voluntary, confidential, and accessible only to those who were provided a link. A standardized email which contained the survey link, and a description of the study was sent to each member of the listserv. All participants received two standardized follow-up emails. Completion of the survey signified a respondent’s consent to participate. No direct benefits or compensation were offered for participation in the study. The study was determined to be exempt from the human subjects review by the University of California, San Francisco, Institutional Review Board.

### 2.2. Survey Instrument

The survey was developed with a multidisciplinary team of psychiatrists, nurses, neonatologists, and a NICU medical director. Feedback was provided by former NICU caregivers to ensure family priorities were addressed in the survey. The 25-item instrument used multiple-choice selection and Likert type rating scales and included information about hospital type, patient volume, NICU level of care, and NICU-based psychosocial support staffing. The survey included a series of questions regarding mental health screening and services, including staff involved in screening/treatment, screening instruments and treatment modalities, and a timeline for screening. Additional questions probed respondents’ perceived challenges and barriers to implementing mental health screening and services in the NICU (survey available upon request).

### 2.3. Data Analysis

Survey responses were collected and maintained in REDCap and imported into Microsoft Excel for cleaning. Cleaned data was then imported into GraphPad Prism version 8.0 (GraphPad Software, La Jolla California USA, www.graphpad.com, accessed on 27 May 2022) for analysis. Descriptive statistics were calculated for all variables. Sites were then grouped according to hospital type (community vs. academic), NICU level (I-III vs. IV), monthly patient volume (0–50 vs. 51+), and approach to PMAD screening (routine vs. non-routine). Comparative analyses using Fisher’s exact test were then performed to evaluate differences across these groups. *p*-values of <0.05 were considered statistically significant.

## 3. Results

### 3.1. Participants

Completed survey responses were obtained from 75 out of 700 potential sites (response rate = 10.7%). These NICUs represented a diversity of geographic regions, hospital types, NICU levels, and patient volumes (Table 1).

Compared to the available national directories [23,25], our sample contained a relative overrepresentation of academic, children’s hospital-based, and level IV NICUs (Table 2).

### 3.2. Psychosocial Staffing

Most NICUs had a social worker as a member of staff (*n* = 71, 95%). The prevalence of other NICU-based psychosocial support staff were child-life specialists (*n* = 21, 28%), psychologists (*n* = 8, 11%), mental health nurse specialists (*n* = 4, 5%), psychiatrists (*n* = 2, 3%), pastoral/spiritual care (*n* = 3, 4%), or palliative care (*n* = 2, 3%). If psychosocial support staff were not core NICU staff, some NICUs had access to these providers on a consultation basis (*n* = 56, 75%). The prevalence of mental health consultants from the hospital to the NICU were psychiatrists (*n* = 35, 47%), psychologists (*n* = 28, 38%), child-life specialists (*n* = 23, 31%), and mental health nurse specialists (*n* = 10, 14%). Over half of the NICUs had no access to a psychiatrist (*n* = 40, 53%) or psychologist (*n* = 43, 57%), either on staff or as a consultant. Child-life specialists were more likely to be a part of NICU staff at academic vs. community (51% vs. 3%; *p* < 0.001), level IV vs. level I-III (68% vs. 11%; *p* < 0.001), and high-volume vs. low-volume (43% vs. 13%; *p* = 0.047) programs.

### 3.3. Caregiver Education

Less than half of the NICUs routinely provided caregivers with psychoeducation about mental health self-care and services related to PMADs (*n* = 35, 47%). In NICUs that did provide routine mental health education, this was most often delivered via one-on-one interactions with the caregiver (*n* = 31, 89%), or through educational brochures (*n* = 20, 57%), group classes (*n* = 9, 26%), mobile apps or other virtual media (*n* = 2, 6%). Psychoeducation was provided by a social worker (*n* = 30, 86%), registered nurse (*n* = 20, 57%), physician (*n* = 16, 46%), psychologist (*n* = 11, 31%), child-life specialist (*n* = 4, 11%), or mental health nurse specialist (*n* = 1, 3%). There were no significant differences in the provision of psychoeducation by geographic region, hospital type, NICU level or patient volume (Table 3).

### 3.4. Caregiver PMAD Screening

Less than half of the NICUs reported routine screening of caregivers for PPD or other mental health disorders (*n* = 33, 44%) (Table 3). Nearly one-quarter of the NICUs did not provide any screening (*n* = 18, 24%). Community-based NICUs were more likely to fail to provide routine screening when compared to NICUs in academic medical centers (39% vs. 8%; *p* = 0.01). In NICUs that did conduct routine screening, mothers of hospitalized infants were always screened for one or more PMADs (*n* = 33, 100%), whereas screening was provided less often to fathers (*n* = 15, 45%) or other caregivers (*n* = 3, 9%). Screening was conducted by a social worker (*n* = 25, 76%), nurse (*n* = 14, 42%), physician (*n* = 3, 9%) or psychologist (*n* = 3, 9%). Screening was administered at various time points following admission to the NICU, including within 48 h (*n* = 9, 27%), within the first week (*n* = 6, 18%), within the first month (*n* = 6, 18%), or at multiple time points (*n* = 12, 37%). PMAD screening instruments used included the Edinburgh Postnatal Depression Scale (EPDS) (*n* = 19, 58%), Postpartum Depression Screening Scale (PDSS) (*n* = 3, 9%), Patient Health Questionnaire (PHQ-9) (*n* = 3, 9%), Impact of Event Scale (IES-R) (*n* = 1, 3%), Center for Epidemiological Studies-Depression (CES-D) (*n* = 1, 3%), Perinatal Anxiety Screening Scale (PASS) (*n* = 1, 3%) or the instrument was unknown to the respondent (*n* = 8, 24%).

NICUs that routinely conducted PMAD screening with caregivers were more likely than non-routine screening NICUs to screen at multiple defined time points (37% vs. 5%; *p* = 0.009), screen using the EPDS (58% vs. 22%; *p* = 0.02), screen fathers in addition to mothers (45% vs. 18%; *p* = 0.046), and routinely provide psychoeducation to parents and caregivers (79% vs. 21%; *p* < 0.001). NICUs with routine PMAD screening were also more likely to have access to a psychiatrist (73% vs. 26%; *p* < 0.001) or psychologist (52% vs. 26%; *p* = 0.03) via consultation.

### 3.5. Referral and Treatment

If a caregiver was identified to have PMAD symptoms, a majority of the participating NICUs provided referrals for outpatient treatment (*n* = 57, 76%) (Table 4). Half of NICUs provided some psychosocial support services within the NICU (*n* = 40, 53%), most often caregiver support groups (Table 4).

There were no significant differences in provision of psychosocial support services by geographic region, hospital type, NICU level, or patient volume (Table 3). NICUs that routinely screened caregivers for PMADs were significantly more likely to provide referrals for treatment than NICUs that did not conduct routine caregiver PMAD screening (91% vs. 64%; *p* = 0.01).

### 3.6. Challenges in Implementing NICU Caregiver Mental Health Services

Only 22% of respondents agreed that caregivers in their NICU received comprehensive “state of the art” psychosocial care, and only 30% agreed that the staff available for caregiver psychosocial support in their NICU was adequate for their needs. The most commonly cited barriers to implementing psychosocial care in the NICU were related to pragmatic concerns, specifically insufficient funding for psychosocial staff (72%), reimbursement (38%), and lack of time to provide care (36%). Other issues, such as the support of the medical team (18%), a lack of evidence-based practices (15%), and the importance of psychosocial care (12%) were less often seen as challenges (Figure 1).

Figure 1 Participants rated potential barriers to psychosocial care on a scale of 0–3, with 0 indicating “not a barrier” and 3 indicating a “major barrier”. Data shown are the mean ± 95% confidence interval for each category. The dashed vertical line represents a rating of 2, corresponding with a “moderate barrier”.

Compared to NICUs that did not conduct routine PMAD screening, NICUs that had routine screening were significantly less likely to perceive barriers associated with time (16% vs. 53%; *p* = 0.002), medical team or social work support (6% vs. 26%; *p* = 0.03), or a lack of evidence-based psychosocial approaches (3% vs. 21%; *p* = 0.03).

## 4. Discussion

To our knowledge, this survey provides the first national data on routine mental health screening and treatment programs for primary caregivers in NICUs. Our findings are consistent with a recent systematic review, which found few studies describing universal screening for NICU caregivers, despite consensus agreement that postpartum psychosocial care is essential [22]. While some hospitals have established programs to address PMADs in parents of newborn infants, this study shows that widespread adoption of NICU-based mental health screening remains inadequate. For NICUs that have implemented routine screening, variation exists with regard to the staff involved, timing of screening, instruments used, and treatment availability. With growing US legislation calling for standardized PPD screening, mental health screening guidelines across inpatient specialty units, such as NICUs, are strongly recommended [26,27].

### 4.1. Staff

Consistent with staffing mandates enacted by many states [28], most NICUs had a social worker as a psychosocial support staff. However, we found that few NICUs met the NPA’s recommendations for employing at least one doctoral level psychologist in NICUs with 20 or more beds [21]. Access to a clinical psychologist has contributed to the success of screening and referral programs in primary care and NICU settings [22] and, indeed, our study found that access to a psychiatrist or psychologist was more common in NICUs providing universal screening.

We found that the majority of screening was conducted by either NICU social workers or nursing staff. This is consistent with previous studies reporting the utilization of various professional staff members to conduct the initial screening, including nursing staff [17,29,30,31], lactation consultants [29,31], social workers [27,30], case managers [30], and psychology fellows [27]. In order to maximize screening rates and detection, PMAD screening should be multidisciplinary and integrated into preexisting workflows with clearly delineated roles for each provider [29,32,33].

### 4.2. Education

Despite recommendations that all NICU caregivers be provided with education about mental health self-care and services [21,26,27], we found that less than half of the NICU’s surveyed provided routine psychoeducation. Routine provision of psychoeducation was more common at NICUs that also provided routine screening, which may reflect the integration of education into the screening protocol. For example, following screening, Lambarth and Green 2015 provided mothers that had a negative screen with information on self-care and community resources such as Baby Blues Connection and their local chapter of Post-Partum Support International [32]. Alternatively, Moyera et al. 2021 coupled screening and psychoeducation into 30 min consultations between each parent and either a social worker or psychologist [27]. While we did not collect information regarding the content of the psychoeducation provided in respondent NICUs, the most common means of providing support was via one-on-one interaction with a social worker or nurse, or via informational brochures. As nearly all NICUs included a social worker on their staff, this approach may represent a cost-effective means for providing psychoeducation without the need for hiring additional NMHPs.

### 4.3. Screening Timing and Instruments

Previous studies identified PMADs through screening at various time points including: 1–3 days [17], 2 weeks [27,29], and 1 month [31], often using multiple time points [30,32]. Although we collected information on time frames rather than specific time points, provision of routine screening was similarly found to range between days (e.g., within 48 h, within the first week) and several weeks (e.g., within one month), with many sites also indicating multiple time points for screening. This variation is consistent with the ongoing debate around the optimal timing to administer PMAD screening in the NICU. Hynan et al. has recommended that screening should be carried out within the first week in order to evaluate parents whose babies are in the NICU for only a few days, as parents with short stays have reported high levels of emotional distress [21]. Conversely, others argue that screening be delayed until two weeks post admission, as early screening may instead capture parents with an acute adjustment reaction that does not persist over time, and for whom mental health interventions are not indicated [27,33].

Alternatively, a tiered screening approach may provide the most effective use of limited resources. There is some evidence that the EPDS administered within 4 days postpartum is predictive for the diagnosis of PPD in mothers at 4–6 weeks [34,35,36]. Parents who score low at the initial screening may be excluded from additional screening or evaluation. However, because the prevalence of PPD appears to peak between 2 to 6 months after delivery [37], repeated screening may be appropriate in cases of prolonged hospitalization. Parents who have known risk factors for the development of PMADs such as fetal anomalies, preterm delivery, or psychiatric history will also require ongoing screening and follow-up [21,27,36].

We also discovered notable variation in the use of screening instruments among sites. NICUs that performed routine screening were more likely to screen for PPD using the EPDS, while non-routine screening sites most often used the PDSS. Although both the PDSS and EPDS are well-validated in NICU mothers, only the EPDS has been validated in fathers [38,39]. The longer length of the PDSS may also be a barrier for some populations, especially non-English speakers [22]. Alternatively, serial testing with a two-question screen from the USPSTF, followed by a second more specific instrument for those who have a positive result, may be a reasonable strategy to reduce false positives while minimizing false negatives [40]. Given that Black, Indigenous and people of color in the US experience both higher rates of PPD and significant treatment disparities [41,42], an anti-oppressive lens should be used in selecting screening instruments that are appropriate for the populations served by the NICU. Furthermore, although the literature emphasizes the importance of screening for anxiety and trauma symptoms in addition to PPD [19], only two NICUs reported using validated instruments to screen for symptoms of PTSD or anxiety. While it is important to evaluate parents and caregivers for all forms of emotional and psychological distress, the use of multiple instruments may result in screening fatigue. Additional research is needed to determine feasibility and patient acceptance and to optimize the screening protocol for accurate and timely detection of PMADs in the NICU.

### 4.4. Referral and Treatment

While screening is important for the detection of PMADs, screening alone is insufficient to improve clinical outcomes and must be coupled with appropriate follow-up assessment and treatment by a mental health care provider when indicated [21,43]. It is suggested that counseling and therapy take place within the NICU, as outside referrals are often not accomplished due to significant competing demands and insurance constraints [22,29]. Ideally, this care would be delivered at the bedside, since many parents with ill children will not leave their child [44]. Unfortunately, we found that relatively few NICUs offered treatment services in the NICU itself. This is likely a reflection of the low prevalence of NMHPs on staff in NICUs, as even among routine screening sites, less than 25% employed a staff psychologist, psychiatrist, or mental-health nurse specialist. Future research should examine the feasibility of leveraging inpatient psychiatry consultants to deliver treatment for caregivers requiring medication at the bedside. An alternative to in-person therapy may be the expanded use of telemedicine, a promising approach for both screening [45] and treatment of PPD [46,47].

### 4.5. Challenges and Barriers

Pragmatic concerns such as insufficient funding for NICU psychosocial staff, lack of time, and challenges with reimbursement were the most commonly cited barriers to providing psychosocial care. These findings are consistent with previous studies that found that lack of in-house psychiatric services, time constraints, and absence of appropriate billing codes are common barriers to implementing NICU mental health care programs [22,27]. The overall similarities in barriers faced by NICUs across hospital type indicate that a more systematic approach may be needed to encourage the expansion of available psychosocial funding and services. At an administrative level, this will likely require additional research into the cost-effectiveness of providing NICU psychosocial care [48].

In contrast to respondents from NICUs that provided universal screening, NICUs that did not provide regular universal screening were more likely to perceive significant barriers related to conceptual issues such as a lack of support from social work or the medical team. This speaks to the importance of multidisciplinary collaboration when designing PMAD screening programs in the NICU. Both staff engagement during program development and the assignment of program champions who have protected administrative time have been found to be essential for the implementation and sustainability of programs [22,49].

Finally, we found that academic centers were more likely than community NICUs to perceive lack of time as a significant barrier. Academic centers in our sample were also significantly more likely to be level IV and high-volume NICUs. The larger patient volume and greater complexity of care at these sites increases both the number of parents who require screening and competing time demands for existing team members. Universal screening programs may be more easily implemented in smaller NICUs who are less likely to require additional staff dedicated to screening and referral [19].

### 4.6. Limitations

There are important limitations to consider when interpreting the results from this survey. Due to the low response rate (10.7%), it is possible that the sample is not representative of NICUs nationwide. While no comprehensive directory of NICUs in the United States currently exists, two large directories are available [23,25]. When compared to these national directories, this study contained an overrepresentation of academic, northeastern, and level IV NICUs, thus the results may not be generalizable (Table 2).

Given the low response rate, voluntary response and non-response biases likely created an overrepresentation of respondents from NICUs that provide PMAD screening and who were motivated to share their practices. In addition, social desirability bias may have influenced respondents to provide optimistic estimates of the screening practices in their NICU. Therefore, the true national prevalence of routine PMAD screening is likely lower than that reported in our survey. Future studies involving medical records review will be needed to determine prevalence with greater precision.

## 5. Conclusions

This was the first study to investigate the national prevalence of routine mental health screening for parents and other primary caregivers in the NICU. We found that less than half of NICUs routinely screened parents for PPD, and nearly one-quarter of the NICUs did not screen at all. Where routine screening was provided, significant variation existed in the timing of screening and instruments used. Fathers and other primary caregivers were frequently excluded from screening, and rarely underwent site screening for symptoms of anxiety or PTSD. While the presence of mental health professionals such as psychiatrists, psychologists and mental health nurse practitioners appears to be an important facilitator for implementing screening, significant barriers related to reimbursement and a lack of funding limits their broad availability. Future studies should focus on how individual NICUs have successfully established routine mental health screening and treatment, the specific processes involved, and how these can be scaled and sustained to support the health and well-being of caregivers and their infants.

## Figures and Tables

**Figure 1 children-09-00793-f001:**
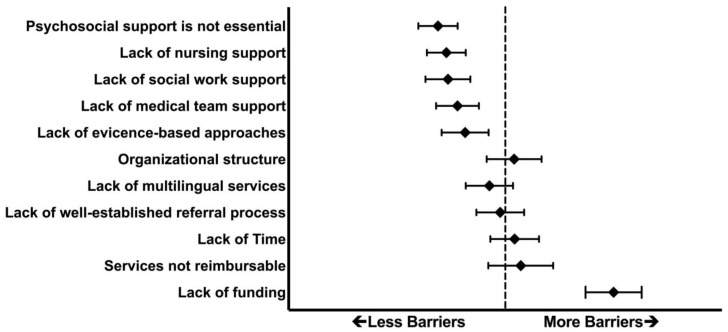
Challenges and Barriers to Caregiver Psychosocial Care in the NICU.

**Table 1 children-09-00793-t001:** Characteristics of participating NICU programs (*n* = 75).

Geographic Distribution (US Census Regions) ^a^	*n* (%)
Midwest	15 (20%)
West	18 (24%)
South	18 (24%)
Northeast	24 (32%)
Type of NICU Setting ^b^	
Independent Children’s Hospital vs. Not	
Children’s hospital	28 (37%)
Non-children’s hospital	47 (63%)
Academic vs. Community Hospital	
Academic medical center	25 (33%)
Community center	32 (42%)
NICU Level	
I	1 (1%)
II	7 (9%)
III	45 (60%)
IV	22 (29%)
Patient Volume (per month)	
0–50	38 (50%)
51–100	24 (32%)
101–150	6 (8%)
150–200	2 (3%)
200+	5 (7%)

^a^ The following states had one or more programs represented in the study (*n* = 31): AR, CA, CT, DC, FL, GA, HI, IA, IL, IN, KY, LA, MA, MD, MI, NC, NE, NJ, NM, NY, OH, OR, PA, RI, SD, TN, TX, UT, VA, WA, WI. ^b^ Participants could select >1 option for NICU setting.

**Table 2 children-09-00793-t002:** Characteristics of participating NICU programs (*n* = 75).

Geographic Distribution (US Census Regions)	Study Sample *n* (%)	National Sample *n* (%) ^a^	*p*-Value
Midwest	15 (20%)	303 (21.8%)	0.78
West	18 (24%)	323 (23.3%)	0.89
South	18 (24%)	534 (38.5%)	0.01
Northeast	24 (32%)	229 (16.5%)	0.001
Type of NICU Setting ^b^			
Independent Children’s Hospital vs. Not			
Children’s hospital	28 (37%)	212 (15.3%)	<0.001
Non-children’s hospital	47 (63%)	1175 (84.7%)	<0.001
Academic vs. Community Hospital			
Academic medical center	25 (33%)	162 (12.6%)	<0.001
Community center	32 (42%)	1128 (87.4%)	<0.001
NICU Level			
I	1 (1%)	0 (0%)	0.05
II	7 (9%)	552 (39.8%)	<0.001
III	45 (60%)	708 (51.1%)	0.15
IV	22 (29%)	127 (9.2%)	<0.001
Patient Volume (per month) ^c^			
0–50	38 (50%)	426 (41.4%)	0.15
51–100	24 (32%)	344 (33.5%)	0.89
101–150	6 (8%)	157 (15.3%)	0.09
150–200	2 (3%)	61 (5.9%)	0.31
200+	5 (7%)	40 (3.9%)	0.23

^a^ National data on geographic distribution, NICU setting, and NICU level are based on the Neonatology Solutions database (*n* = 1387). Patient volume data is based off of the AAP database (*n* = 1028). ^b^ Participants could select >1 option for NICU setting. ^c^ The AAP national directory contained information on bed number rather than patient volume. For the national data, the following groupings are used: 0–19 beds, 20–39 beds, 40–59 beds, 60–79 beds, 80+ beds.

**Table 3 children-09-00793-t003:** Provision of routine mental health education, screening, and treatment, by NICU characteristics.

	Education ^a^ *n* (%)	Screening ^b^ *n* (%)	Treatment *n* (%)
Geographic Region			
Midwest (*n* = 15)	8 (53%)	8 (83%)	12 (80%)
West (*n* = 18)	8 (44%)	10 (56%)	14 (78%)
South (*n* = 18)	9 (50%)	9 (50%)	14 (78%)
Northeast (*n* = 24)	10 (42%)	6 (25%)	20 (83%)
Hospital Type ^c^			
Independent Children’s Hospital vs. Not			
Children’s hospital (*n* = 28)	15 (54%)	14 (50%)	23 (82%)
Non-children’s hospital (*n* = 47)	20 (43%)	19 (40%)	37 (79%)
Academic vs. Community Hospital			
Academic (*n* = 39)	21 (54%)	20 (51%)	32 (82%)
Community (*n* = 36)	14 (39%)	13 (36%)	28 (78%)
NICU Level			
I–III (*n* = 53)	22 (42%)	20 (38%)	42 (79%)
IV (*n* = 22)	13 (59%)	13 (59%)	18 (82%)
Patient Volume (per month)			
0–50 (*n* = 38)	16 (42%)	14 (37%)	30 (79%)
51+ (*n* = 37)	19 (51%)	19 (51%)	30 (81%)

^a^ A site was categorized as providing “routine mental health education” if they reported “always” or “usually” providing psychoeducation and indicated which types of psychoeducation were provided. ^b^ A site was categorized as providing “routine mental health screening” if the site reported “always” or “usually” providing screening and indicated specific time points for screening. ^c^ Participants could select >1 option for NICU setting. There was no overlap between academic and community centers. A subset of programs indicated only whether the hospital was a children’s hospital. In these cases, the hospital name was used to investigate and sort the remaining programs into academic or community for comparative analysis.

**Table 4 children-09-00793-t004:** Mental health treatment referrals and services offered (*n* = 60).

	*n* (%)
Referrals	
Referral to therapist in the community	39 (65%)
Referral to therapist in hospital system	34 (57%)
Referral to psychiatrist in hospital system	29 (48%)
Referral to psychiatrist in the community	27 (45%)
Services	
Support groups	32 (53%)
Individual supportive psychotherapy	19 (32%)
Couples/Marital counseling	6 (10%)
Cognitive behavioral therapy	4 (7%)
Family therapy	3 (5%)
Problem solving skills training (PSST)	2 (3%)

## Data Availability

The datasets used and/or analyzed during the current study are available from the corresponding author on reasonable request.

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
