# Peer review of "National Cross-Sectional Study of Mental Health Screening Practices for Primary Caregivers of NICU Infants"

_children, 2022, doi:10.3390/children9060793_

Round 1

Reviewer 1 Report

Thank you for inviting me to review this manuscript. I carefully read it and I found it very interesting and well-written. However, I suggest the authors to focus more on the discussion section, thus avoiding to merely repeat the results. To my opinion, sub-paragraphs are not necessary since the authors should interpret results, in a detailed but holistic approach, place them in the context of previous research, explaining what they mean  for future research, as well as for possible real-life applications.

Reviewer 2 Report

First of all, thank you for the opportunity to review this manuscript. The topic addressed is interesting, and little addressed from the practical assistance in the day to day of the clinic as well as from the investigation. Therefore, the theme is very suitable. In addition, the article is well written, makes it easy to read. The goal setting is correct. The results are expressed clearly. In the discussion, the limitations are addressed. Despite this, the article could improve by addressing a few minor issues that are detailed below:

-A multivariable analysis is missing

--It would be appropriate to make a more analytical discussion and not only descriptive

--A limitation that must be addressed is having used the online questionnaire. This may limit accessibility to certain social groups. Could this have influenced the results?

-The bibliographical references are not correctly expressed. In addition, it would be convenient to update the bibliographic references, most of them are older than 10 years
